# Dynamic Pattern Alignment Learning for Pre-training Lightweight Human-Centric Vision Models

## Abstract

Human-centric vision models (HVMs) have achieved remarkable generalization due to large-scale pretraining on massive person images. However, their dependence on large neural architectures and the restricted accessibility of pretraining data significantly limits their practicality in real-world applications. To address this limitation, we propose Dynamic Pattern Alignment Learning (DPAL), a distillation-based pretraining framework that effectively transfers generalization capability of large HVMs to lightweight ones by mimicking three heterogeneous visual patterns (i.e., global identity pattern, local shape pattern and multi-person interaction pattern). Specifically, we design a dynamic pattern decoder (D-PaDe) that functions as a dynamic mixture of expert (Dy-MoE) model with three specialized experts. This design allows each visual pattern to be generated independently, thus avoiding optimization conflicts caused by pattern heterogeneity during training. Moreover, three alignment objectives are designed to narrow the visual representation gap between large HVMs and lightweight ones at global image, local pixel, and instance relation levels, respectively. Once pretrained, the lightweight model acquires strong generalization capability from large HVMs, thereby supporting a wide range of human-centric vision tasks. Extensive experiments conducted on 15 challenging datasets demonstrate the effectiveness of the DPAL. Remarkably, when employing PATH-B as the teacher, DPAL-ViT/Ti (5M parameters) achieves surprising generalizability similar to existing large HVMs such as PATH-B (84M) and Sapiens-L (307M), and outperforms previous distillation-based pretraining methods including Proteus-ViT/Ti (5M) and TinyMiM-ViT/Ti (5M) by a large margin. More importantly, the DPAL is performed on a limited dataset (i.e., around 1M unlabeled images) that is unseen for large HVMs, which bypasses the need for those inaccessible or constrained pretraining datasets, offering an affordable approach to generalizable HVMs. All code and checkpoints will be publicly available[1].

## 1 Introduction

Recent years have witnessed remarkable progresses in human-centric visual perception (HVP) (Yuan et al., 2023; Khirodkar et al., 2024; Chen et al., 2023). This success is mainly attributed to the advancement in pretraining of large vision models with massive collected data. By leveraging such extensive pretraining, large human-centric vision models (HVMs) are able to learn generalizable visual patterns, which widely benefit various human-centric perception tasks, such as single-person discrimination (Fu et al., 2021a; He et al., 2021), dense prediction (Yuan et al., 2021; Li et al., 2020) and multi-person visual understanding (Ci et al., 2023; Tang et al., 2023).

Although large HVMs exhibit strong generalization capability, there are two primary computational challenges that significantly limit the practicality of large HVMs in real-world application. First, large HVMs typically exhibit substantial model size, demanding considerable computational resources and making the pretraining of HVMs prohibitively expensive for most researchers. For example, Sapiens (Khirodkar et al., 2024), a typical self-supervised pretrained HVM, employs ViT-G (2B parameters) as the model architecture and trains it on Humans-300M for 18 days using 1024 A100

---

[1]https://anonymous.4open.science/r/DPAL-23D6

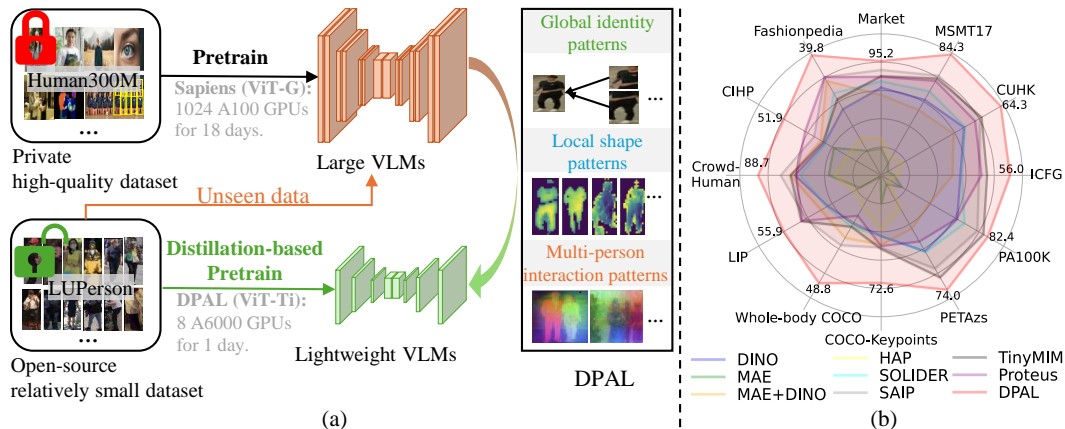

Figure 1: (a) Existing HVMs are limited in real-world application due to two factors: 1) Large model size with heavy computational costs, and 2) high-quality pretraining datasets are strictly constrained. To address these, Dynamic Pattern Alignment Learning (DPAL) is proposed to pretrain lightweight HVMs by distilling generalization capability from large HVMs across three typical human visual patterns. (b) Extensive experiments conducted on 12 datasets demonstrate effectiveness of the DPAL.

GPUs. The PATH, a representative supervised pretrained HVM, trains ViT-L on HumanBench with 11M images using 64 V100 GPUs. This substantial dependence on huge computational resources complicates real-world deployment of HVMs. Second, pretraining HVMs relies on extensive and high-quality datasets. However, the qualified pretraining datasets such as Humans-300M (Khirodkar et al., 2024) and HumanBench (Tang et al., 2023) are often inaccessible or strictly constrained due to the concern about violations to copyright ownership of these digital assets. These challenges pose significant limitation on the broad applicability of existing pretraining methods.

To achieve strong generalizability while maintain the broad applicability, exploiting the potential of lightweight HVMs via extensive pretraining is a promising direction. Previous works (Caron et al., 2021; He et al., 2022) on lightweight HVMs focus on self-supervised pretraining by using alternative dataset such as LUPerson (Fu et al., 2021a), which is widely used for advancements in human-centric vision foundation models. Nevertheless, the scale of publicly available dataset is relatively small, e.g., the number of image in LUPerson is around 4M which is much less than that of those private datasets, limiting the optimization of lightweight HVMs. Motivated by these issues, we focus on a significant question that is considerably less studied: *is it possible to replicate strong generalization capability from large HVMs to lightweight HVMs without requiring access to those inaccessible or strictly constrained pretraining datasets?*

In this work, we seek to answer the question by exploring distillation-based pretraining framework, which transfers the generalization capability of large HVMs to lightweight ones by leveraging limited dataset as a medium. Specifically, human-centric visual perception generally relies on three typical visual patterns: 1) global identity pattern for single-person discrimination tasks such as Person ReID, 2) local shape pattern for dense prediction tasks such as pose estimation, and 3) multi-person interaction pattern for multi-instance visual understanding tasks like pedestrian detection. This reliance suggests that a robust human-centric vision model with strong generalization must be capable of acquiring all three visual patterns. However, directly learning different patterns is hindered by optimization conflicts caused by pattern heterogeneity. As analyzed in previous works (Chen et al., 2023; Yuan et al., 2023), global pattern learning tends to homogenize pixel representations, sacrificing fine-grained information. In contrast, local pattern learning hurts the global identity information, as it is expected to learn semantic-consistent region representations. To overcome these limitations, we propose **D**ynamic **P**attern **A**lignment **L**earning (DPAL), a novel distillation-based pretraining framework that effectively transfers generalization capability of large HVMs to lightweight ones by mimicking those heterogeneous visual patterns. Specifically, we firstly design a dynamic pattern decoder (D-PaDe), acting as a dynamic MoE model with three experts dedicated to separately processing local, global, and relational patterns. It activates only one expert module per input to ensure alignment with one pattern. And then, three alignment objectives are further designed to minimize visual representation gap between large HVMs and lightweight ones at global image, local

pixel, and instance relation levels, respectively. Once pretrained, the lightweight model successfully acquires strong generalization capability of large HVMs, thus supporting various downstream tasks.

We conduct extensive experiments on 15 challenging benchmarks involving 9 representative human-centric visual perception tasks and three cross-domain visual perception tasks, demonstrating the impressive effectiveness of proposed DPAL. Remarkably, when employing PATH-B as the teacher, DPAL-ViT/Ti (5M parameters) achieves surprising generalizability competitive to that of existing large HVMs such as PATH-B (84M) and Sapiens-L (307M), and outperforms previous distillation-based pretraining methods including Proteus-ViT/Ti (5M) and TinyMiM-ViT/Ti (5M) by a large margin. More importantly, the distillation pretraining in DPAL is performed on the limited dataset that is unseen for large HVMs for around 1 day, without the need for those inaccessible or constrained pretraining datasets, offering an affordable approach to generalizable HVMs.

## 2 RELATED WORKS

### 2.1 HUMAN-CENTRIC VISION MODEL

Human-centric vision models (HVMs) refer to pretrained models specifically designed and trained to handle human-related visual tasks, including person re-identification (ReID) (Fu et al., 2022; He et al., 2021; Luo et al., 2021), text-to-image person ReID (Shao et al., 2022; Ding et al., 2021b; Suo et al., 2022; Shao et al., 2023), pedestrian attribute recognition (Jia et al., 2022; 2021), action recognition (Sun et al., 2022; Qian et al., 2024; Zhong et al., 2023), and body structure understanding such as 2D/3D pose estimation (Choi et al., 2022; Xu et al., 2022; Yuan et al., 2021; Li et al., 2020).

Recent works (Yuan et al., 2023; Ci et al., 2023; Tang et al., 2023; Khirodkar et al., 2024; Chen et al., 2023) have proposed HVMs tailored for human-centric tasks. For example, both SOLIDER (Chen et al., 2023) and the HAP (Yuan et al., 2023) propose to leverage the human body layouts for pretraining, demonstrating the importance of human body structure priors in learning robust human-centric visual representations. In a different line, several studies such as PATH (Tang et al., 2023) and Sapiens (Khirodkar et al., 2024) focus on the construction of high-qualified pretraining datasets, resulting in large-scale person-centric data sources such as HumanBench-11M and Humans-300M. Although these methods have achieved impressive results in human-centric downstream tasks, the large parameter size and inaccessible large-scale dataset of HVMs (Yuan et al., 2023; Tang et al., 2023; Ci et al., 2023; Khirodkar et al., 2024) makes them unsuitable for real-world application. Therefore, our primary objective is to leverage knowledge distillation to effectively transfer generalization capability of large HVMs to a lightweight counterpart without relying on source pretraining datasets.

### 2.2 KNOWLEDGE DISTILLATION

Traditional knowledge distillation (Hinton et al., 2015; Chen et al., 2019; Yin et al., 2020; Chen et al., 2022; Yang et al., 2024; Son et al., 2024; Fan et al., 2024) aims to model compression via aligning the outputs from small model to that of larger model. To replicate representation capability of large visual foundation models, which are usually computationally intensive, several studies (Zhang et al., 2025; Ren et al., 2023) study distillation-based pretraining for compact foundation models. For example, the TinyMIM (Ren et al., 2023) uses ImageNet (Deng et al., 2009) to explore the impact of various distillation factors such as aligning objectives and the way of distillation, and find that distilling attentions in vision transformer is the key to narrow the gap between the small MAE model and the large one. Unlike traditional single-stage distillation, G2SD (Huang et al., 2023) proposes a two-stage distillation approach, where a MAE pretrained model is used as the teacher model for distillation, followed by distillation on specific tasks. Furthermore, theia (Shang et al., 2024) proposes a multi-teacher off-the-shelf distillation strategy, which learns rich visual representations of multiple teachers simultaneously. Proteus (Zhang et al., 2025) proposes distillation across three different levels of training objectives for mimicking the teacher's behaviors.

These methods illustrate the considerable potential of knowledge distillation in model compression and efficiency improvement. However, existing methods are limited to a single visual pattern (e.g. DeiT (Touvron et al., 2021) captures global identity patterns, G2SD (Huang et al., 2023) focuses on local patterns), resulting in a significant gap in human-centric visual perception tasks. To overcome this limitation, we introduce DPAL, which dynamically learns three typical patterns from large HVMs to enhance generalizability across diverse downstream tasks.

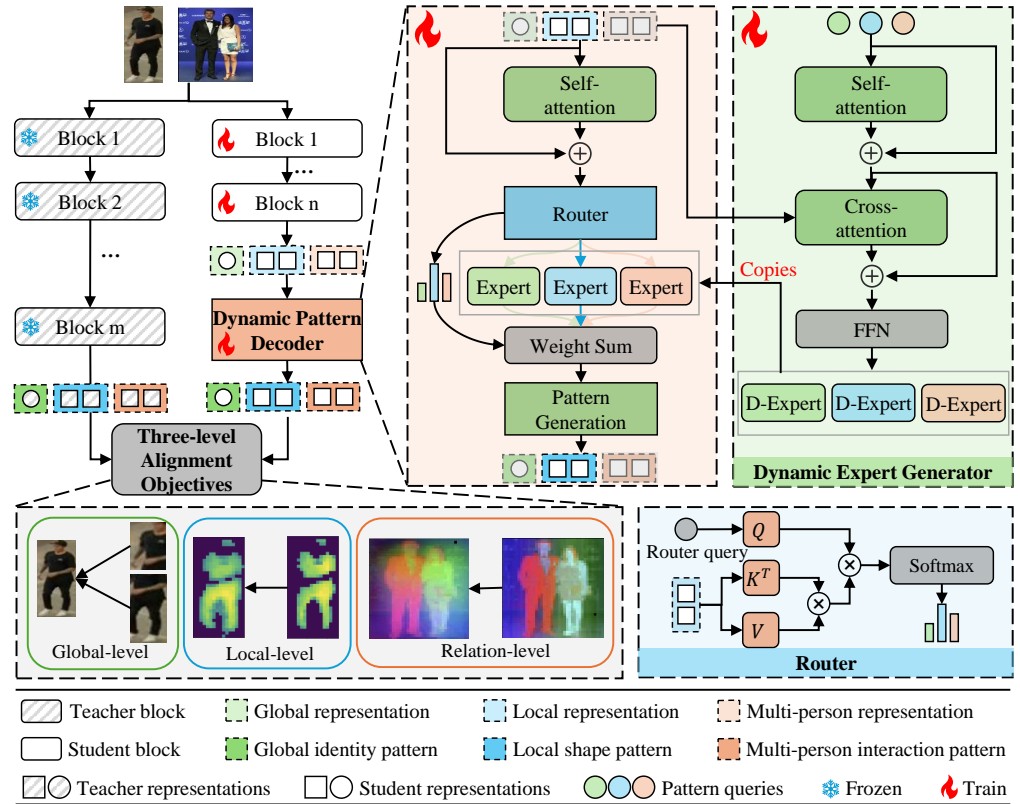

Figure 2: The overview of DPAL. It comprises of (1) Teacher model, (2) Student backbone and (3) Dynamic pattern decoder. Dynamic pattern decoder captures three types of patterns conditioned on the input image and pattern queries. Finally, three-level alignment objectives are employed to acquire generalization capability from large HVMs to lightweight HVMs.

## 3 METHOD

As illustrated in Fig. 2, we propose **D**ynamic **P**attern **A**lignment **L**earning (DPAL), a distillation-based pretraining framework for building generalizable lightweight HVMs. Following standard teacher-student architecture (Caron et al., 2021), the DPAL relies on two major designs: 1) Dynamic Pattern Decoder that extracts three typical human visual patterns in a dynamic way, and 2) three levels of alignment objectives that maximize knowledge transfer from the large HVMs to the lightweight one by leveraging those patterns as the medium.

### 3.1 MODEL ARCHITECTURE

Similar to standard knowledge distillation framework, our distillation-based pretraining framework consists of three major components: visual encoder of student model (s-VisEn), visual encoder of teacher model (t-VisEn) and Dynamic Pattern Decoder (D-PaDe). In particular, the D-PaDe functions as an adapter to align the outputs of s-VisEn with those of t-VisEn by projecting them into a common latent space. Once pretrained, the s-VisEn is retained for supporting downstream tasks, while the t-VisEn and D-PaDe are discarded.

**s-VisEn.** The basic architecture of s-VisEn is built on lightweight vision transformer (e.g., ViT/Ti), which tokenizes input image into numerous vision tokens. Formally, given an input image $I \in \mathbb{R}^{C \times H \times W}$, the s-VisEn is able to output two types of tokens: 1) class token $F_s^g \in \mathbb{R}^{D_s}$ representing $D_s$ dimensional global representation of an image, and 2) patch token $F_s^l \in \mathbb{R}^{L \times D_s}$ representing pixel feature of $L$ image patches.

**t-VisEn.** The architecture of teacher model is a large vision transformer (e.g., ViT/B), and it has been pretrained using large-scale datasets. Therefor, the t-VisEn often has a stronger representation capacity for extracting diverse visual patterns. Unlike to the s-VisEn, a pretrained t-VisEn can be used to yield three types of tokens from an image, including class token $F_t^g \in \mathbb{R}^{D_t}$, patch token $F_t^l \in \mathbb{R}^{L \times D_t}$ and attention token $F_t^r \in \mathbb{R}^{L \times L}$. As demonstrated in previous works (Tang et al., 2023; Ci et al., 2023), class token of a pretrained ViT carries global discriminative visual information, the patch token captures local visual layouts, and intern-patch relations are encoded in attention token. Therefore, the class, patch, and attention tokens from a large pretrained ViT are adopted to serve as the latent representations for global identity, local shape, and interaction patterns, respectively.

**D-PaDe.** To avoid the optimization conflicts caused by pattern heterogeneity, the D-PaDe is designed as a dynamic mixture of experts (MoE) model, which decouples the generation of different patterns. Existing MoE models (Riquelme et al., 2021; Dai et al., 2024) adopt a fix set of small feed-forward network (FFN) as experts, and use a router module to decide which expert should be activated. However, those fixed experts are insufficient to cope with the diversity in visual patterns. To overcome this, we adopt a different strategy, where the experts in D-PaDe are dynamically predicted based on the type of input tokens and pattern queries. Specifically, we denote three learnable tokens as the pattern queries $T_e = [T_e^1, T_e^2, T_e^3]$, each of which is responsible for deciding what expert should be generated. And then, a self-attention module $SA(\cdot)$ is used to encode pattern queries and a cross-attention module $CA(\cdot)$ is further used to adaptively filter the visual tokens with the guide of the pattern queries. Finally, those filtered tokens are projected to computational parameters of three experts $\{E\}_{i=1}^3$ via a linear FFN. Additionally, the router in D-PaDe is typically a linear layer that computes expert selection scores $\{W^e\}_{i=1}^3$ based on the input token type. Based on this, the D-PaDe is activated to decode only one specified pattern once the query is assigned. In the following, we present how to utilize D-PaDe to decode various visual patterns.

## 3.2 PATTERN GENERATION

**Global identity pattern** is generally expressed via global discrimination information. Therefore, this pattern can be directly obtained from single-person image $I_1$, as formulated as follows:

$$\widetilde{F_s^g} = \sum_{i=1}^3 W_i^e \cdot E_i(F_{s1}^g + SA(F_{s1}^g)) \tag{1}$$

where $F_{s1}^g$ is the global token of the single-person image $I_1$, which is extracted from s-VisEn. $W_i^e$ is the weight score for $i$-th expert.

**Local shape pattern** represents local human body shape information. To achieve this shape information, we use the attention score in t-VisEn as the a coarse mask $M_{shape}$, which roughly attends foreground body shape. Then, the local shape pattern is generated by filtering irrelevant patch tokens via the mask. This process can be formulated in Equ. 2:

$$\widetilde{F_s^l} = M_{shape} \cdot \left[ \sum_{i=1}^3 W_i^e \cdot E_i(F_{s1}^l + SA(F_{s1}^l)) \right] \tag{2}$$

where $F_{s1}^l$ is the local patch tokens of the single-person image $I_1$, which are obtained from s-VisEn.

**Multi-person interaction pattern** represents the relational information between different instances. To generate this kind of pattern, we firstly cast image patch tokens $F_{s2}^l$, which are extracted from a multi-person image $I_2$ via the s-VisEn, as latent tokens using D-PaDe. And then, we compute inter-token similarity using softmax nonlinear function. The resulted scores can be viewed as relations among the patches. This process is formulated through Equ. 3:

$$\begin{aligned}
\widetilde{F_{s2}^l} &= \sum_{i=1}^3 W_i^e \cdot E_i(F_{s2}^l + SA(F_{s2}^l)) \\
\widetilde{F_s^r} &= softmax(\frac{\widetilde{F_{s2}^l}^T \widetilde{F_{s2}^l}}{\sqrt{D_s}})
\end{aligned} \tag{3}$$

## 3.3 ALIGNMENT OBJECTIVES

To minimize the generalization gap between lightweight HVMs and large HVMs, we conduct pattern alignment across three different levels, i.e., global image, local pixel, and relation levels.

**Global-level Alignment.** To learn global identity pattern, we construct $M$ multi-view images derived from the single-person image $I_1$. Then, we extract global identity patterns from those multi-view images and minimize their representation gap between the student model and teacher model, as formulated in Equ 4.

$$\ell_g = \frac{1}{M} \sum_{i=1}^{M} \|\widetilde{F_s^g}_i - \widetilde{F_t^g}\|_2 \tag{4}$$

**Local-level Alignment.** To align local shape pattern, we use the MSE loss to encourage the consistency between patch tokens decoded via proposed D-PaDe and local shape pattern extracted from the t-VisEn, as illustrated in Equ 5:

$$\ell_l = \|\widetilde{F_s^l} - \widetilde{F_t^l}\|_2 \tag{5}$$

**Relation-level Alignment.** To further acquire instance-level relationships learned in the teacher model, we enforce the lightweight HVMs mimic the multi-person interaction patterns of large HVMs via KL divergence loss:

$$\ell_r = L_{KL}(\widetilde{F_s^r}, \widetilde{F_t^r}) \tag{6}$$

Three loss functions are combined as the overall learning objective $\mathcal{L}$ for optimizing student model:

$$\mathcal{L} = \lambda_g \ell_g + \lambda_l \ell_l + \lambda_r \ell_r \tag{7}$$

where $\lambda_g$, $\lambda_l$ and $\lambda_r$ are hyperparameters for balancing three learning objectives. For simplicity, we set them to 1 during training.

## 4 EXPERIMENT

### 4.1 EXPERIMENTAL SETTINGS

**Datasets.** Unless otherwise stated, all models in this paper are pretrained on LUP1M, a subset of 1 million single-person images sampled randomly from LUPerson dataset (Fu et al., 2021b). Specificaly, the multi-persons are synthesized by applying a simple copy-paste technique (Ghiasi et al., 2021) to different resolution images. For downstream task evaluation, the pretrained models are exhaustively evaluated on standard benchmark datasets to ensure comprehensive performance assessment. Specifically, we adopt Market1501 (Zheng et al., 2015) and MSMT17 (Wei et al., 2018) for image-to-image ReID (I2I ReID), CUHK-PEDES (Li et al., 2017) and ICFG-PEDES (Ding et al., 2021a) for text-to-image ReID (T2I ReID), PA-100K (Liu et al., 2017) and PETA (Deng et al., 2014) for attribute recognition, COCO-Keypoint (Lin et al., 2014) for pose estimation, Whole-body COCO (Jin et al., 2020) for landmark detection, LIP (Liang et al., 2018) for human parsing, CrowdHuman (Shao et al., 2018) for pedestrain detection, CIHP (Gong et al., 2018) for multiple human parsing, Fashionpedia (Jia et al., 2020) for part-level attribute parsing. In addition to general human-centric visual perception tasks, we further evaluate generalizability of pretrained models using a cross-domain setting, where models are pretrained using natural person images, and fine-tuned on cross-style or cross-species visual perception tasks using images from unseen domains. Therefore, we consider three representative datasets: 1) Humanart (Ju et al., 2023) with person images in unseen styles such as cartoons and sketches; 2) Chimpact-Pose (Ma et al., 2023) with chimpanzee images, and 3) AP-10K (Yu et al., 2021) with common animal images.

**Evaluation.** Following previous works, we adopt $Rank1$ for I2I ReID and T2I ReID, mean accuracy ($mAP$) for attribute recognition, average precision ($AP$) and recall ($AR$) for pose estimation and landmark detection, mean intersection of union ($mIoU$) and mean pixel accuracy ($mAcc$) for human parsing, $AP$ and missing rate ($MR$) for pedestrain detection, $mIoU$ and $AP_p$ for multiple human parsing, $AP_{IoU+F_1}^{box}$ and $AP_{IoU+F_1}^{segm}$ for part-level attribute parsing. To fairly evaluate the pretrained HVMs, any decoder or alignment module (e.g., P-DaDe) is discarded and only pretrained backbone model using different pretraining paradigms (self-supervised/distillation-based) is retained for downstream tasks. For additional details, we refer readers to **Appendix**.

Table 1: Comparison with self-supervised pretraining and distillation-based pretraining methods across three single-person discriminative tasks and three single-person dense prediction tasks. *: Swin-tiny is adopted as the student backbone.

| Method | I2I Person ReID (Market / MSMT) | T2I Person ReID (CUHK / ICFG) | Attribute Recognition (PA100 / PETA) | Pose Estimation ($AP$ / $AR$) | Landmark Detection ($AP$ / $AR$) | Human Parsing ($mIoU$ / $mAcc$) |
|---|---|---|---|---|---|---|
| DINO | 90.5/65.8 | 55.3/40.2 | 77.4/69.3 | 69.3/72.6 | 43.9/57.1 | 48.7/59.3 |
| MAE | 79.7/39.9 | 36.6/19.1 | 68.3/61.1 | 67.0/70.6 | 40.1/52.7 | 43.5/54.0 |
| MAE+DINO | 89.2/61.6 | 52.7/37.7 | 72.9/66.5 | 69.9/73.2 | 45.1/58.2 | 49.8/60.3 |
| HAP | 81.6/42.4 | 40.2/20.4 | 66.3/64.1 | 68.8/72.3 | 42.6/55.4 | 44.4/54.5 |
| SOLIDER | 91.6/69.2 | 55.5/40.7 | 78.6/69.4 | 69.3/72.6 | 44.2/57.2 | 48.9/59.2 |
| SAIP | 93.6/75.6 | 59.2/46.1 | 80.7/71.4 | 70.1/73.3 | 45.7/58.7 | 52.3/63.3 |
| ViTKD | 88.3/65.7 | 46.3/35.5 | 77.6/67.0 | 71.3/74.5 | 44.4/57.3 | 52.0/62.8 |
| MaskedKD | 87.9/62.1 | 52.4/42.8 | 75.6/67.6 | 67.9/71.5 | 40.4/54.0 | 50.9/61.7 |
| ScaleKD | 90.7/68.9 | 57.4/46.5 | 78.2/69.3 | 70.8/74.1 | 43.8/57.0 | 55.6/66.5 |
| TinyMIM | 92.5/74.5 | 59.6/48.7 | 81.7/72.5 | 70.2/73.6 | 44.2/57.5 | 53.0/63.7 |
| Proteus | 92.4/73.5 | 58.0/46.9 | 77.3/68.1 | 70.0/73.3 | 43.3/56.4 | 52.9/63.9 |
| **DPAL** (Ours) | **95.2/84.3** | **64.3/56.0** | **82.4/74.0** | **72.6/75.8** | **48.8/61.5** | **55.9/66.7** |
| **DPAL*** (Ours) | **96.4/86.2** | **66.9/58.5** | **83.1/74.9** | **75.1/78.1** | **53.9/65.7** | **59.3/69.7** |

Table 2: Comparison with state-of-the-arts across three multi-person perception tasks, and three cross-domain visual perception tasks. *: Swin-tiny is adopted as the student backbone.

| Method | Pedestrian Detection ($AP$ / $MR$) | Multiple Human Parsing ($mIoU$ / $AP_p$) | Part Attribute Parsing ($AP^b_{F_1}$ / $AP^m_{F_1}$) | Human Art Estimation ($AP$ / $AR$) | Chimpanzee Estimation ($AP$ / $AR$) | Animal Pose Estimation ($AP$ / $AR$) |
|---|---|---|---|---|---|---|
| DINO | 86.1/51.6 | 46.9/45.7 | 35.4/32.9 | 65.7/69.7 | 16.1/19.2 | 58.0/61.6 |
| MAE | 83.7/56.5 | 45.8/44.4 | 32.0/30.3 | 65.1/69.0 | 13.9/16.7 | 48.8/53.5 |
| MAE+DINO | 86.4/50.2 | 47.4/46.2 | 37.7/35.2 | 67.4/71.1 | 18.2/21.2 | 59.5/63.6 |
| HAP | 83.3/57.5 | 44.1/42.8 | 33.0/30.4 | 66.0/70.0 | 13.2/15.8 | 50.5/54.7 |
| SOLIDER | 85.7/51.9 | 46.8/45.7 | 36.7/34.2 | 66.7/70.6 | 16.0/18.9 | 57.4/61.4 |
| SAIP | 87.1/49.6 | 48.2/46.9 | 38.0/35.5 | 67.5/71.2 | 18.2/21.3 | 60.8/64.6 |
| ViTKD | 86.6/50.3 | 44.1/42.9 | 35.6/33.4 | 68.2/71.8 | 18.9/22.4 | 44.0/48.7 |
| MaskedKD | 84.2/53.5 | 47.6/46.5 | 33.3/31.0 | 64.9/68.9 | 16.2/19.1 | 55.3/59.0 |
| ScaleKD | 87.4/47.5 | 49.0/48.3 | 37.5/34.9 | 65.7/69.6 | 20.8/24.3 | 62.9/66.4 |
| TinyMIM | 86.4/50.3 | 47.1/46.0 | 36.2/33.5 | 67.3/71.0 | 17.5/20.5 | 61.6/65.1 |
| Proteus | 86.0/50.3 | 49.3/47.9 | 38.0/35.4 | 69.4/73.0 | 19.7/23.2 | 64.7/68.1 |
| **DPAL** (Ours) | **88.7/45.5** | **51.9/50.3** | **39.8/37.0** | **69.9/73.4** | **21.9/25.5** | **67.0/70.3** |
| **DPAL*** (Ours) | **90.2/42.3** | **55.8/53.3** | **42.9/39.8** | **72.9/76.1** | **25.2/29.4** | **69.4/73.0** |

## 4.2 COMPARISON WITH STATE-OF-THE-ARTS

In this section, we compare DPAL with existing pretraining methods across a wide range of downstream tasks. Specifically, we conduct a comprehensive evaluation against six self-supervised pretraining paradigms, including DINO (Caron et al., 2021), MAE (He et al., 2022), MAE+DINO (Park et al., 2023), HAP (Yuan et al., 2023), SOLIDER (Chen et al., 2023) and SAIP (Wang et al., 2025). Besides, five distillation-based pretraining paradigms are also included for comprehensive comparison, involving ViTKD (Yang et al., 2024), MaskedKD (Son et al., 2024), ScaleKD (Fan et al., 2024), TinyMIM (Ren et al., 2023) and Proteus (Zhang et al., 2025). We adopt PATH (Tang et al., 2023) as the teacher model and use ViT-Tiny and Swin-Tiny (Liu et al., 2021a) as the student model. All student models are trained using LUP1M dataset for 100 epochs by default.

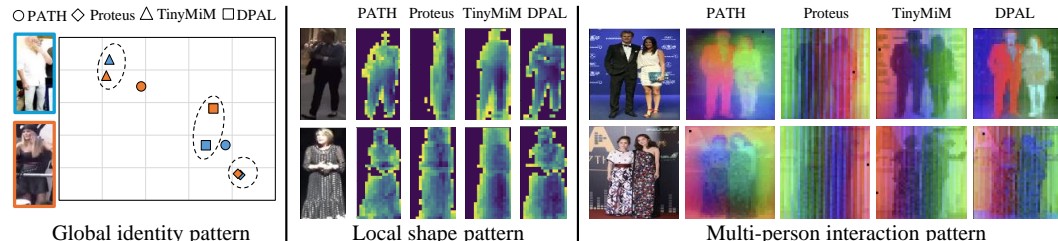

Figure 3: Qualitative comparison between proposed DPAL and state-of-the-art methods.

**Single-person Discriminative Tasks.** We evaluate DPAL's performance on single-person discriminative tasks, including T2I ReID, I2I ReID, and attribute recognition. As shown in Table 1, DPAL outperforms SAIP by significant margins of 1.6% and 8.7% on Market-1501 and MSMT17 for I2I ReID, and by 5.1% and 9.9% on CUHK and ICFG for T2I ReID. For attribute recognition, DPAL outperforms TinyMIM by 0.7% and 1.5% on PA100K and PETAzs, respectively. Additionally, when using Swin-Tiny as the student backbone, the performance improves by an average of 1.63% compared to DPAL. The performance of DPAL on a wide range of single-person discriminative tasks demonstrates that our method is able to effectively learn the global identity patterns from teacher while maintaining strong generalization ability.

**Dense Prediction Tasks.** We compare DPAL with existing methods on three dense prediction tasks, as shown in Table 1. For pose estimation, DPAL achieved 72.6% AP and 75.8% AR, outperforming TinyMiM which scores 70.2% and 73.6%, respectively. In landmark detection, DPAL achieved 48.8% AP and 61.5% AR, surpassing TinyMiM by 4.6% and 4%. For human parsing, DPAL achieves $mIoU$ of 55.9, exceeding TinyMiM by 2.9%. It also achieves 66.7% mean accuracy ($mAcc$), outperforming Proteus by 2.8%. Most notably, employing Swin-Tiny as the student model demonstrates an average 3.42% performance improvement over DPAL. The results substantiate that DPAL successfully learns both global identity patterns and local shape patterns.

**Multi-person Visual Understanding Tasks.** We evaluate DPAL's performance on three multi-person visual understanding tasks, as shown in Table 2, In pedestrian detection, DPAL achieves an AP of 88.7% and an MR of 45.5%, outperforming SAIP (2nd best) by 1.6% and 4.1%, respectively. In multiple human parsing, DPAL achieves a $mIoU$ of 51.9 and an $AP_p$ of 50.3%, surpassing Proteus by 2.6% and 2.4%, respectively. Specifically, using Swin-Tiny as the student model yields an average performance gain of 2.92% relative to DPAL. In part-level attribute parsing, DPAL delivers the best performance across key evaluation metrics.

**Cross-domain Generalization.** As shown in Table 2, the DPAL consistently outperforms previous self-supervised pretraining/distillation-based pretraining methods on two different scenarios unseen in the pretraining phase, including 1 cross-style recognition task and 2 cross-species recognition tasks. This indicates that learning three typical human visual patterns from large HVMs enables generalizable representations for cross-domain adaptation. Similar findings are also observed when applying DPAL to general vision-language tasks (**Appendix.C.3**).

**Qualitative Comparison.** To validate whether DPAL correctly captures the three visual patterns, we visualize the model's outputs. As shown in the figure 3, for global identity patterns, DPAL can distinguish between different instances in the representation space, while both Proteus and TinyMIM struggle to do so. For local shape patterns, DPAL achieves results comparable to those of the teacher model. Additionally, it can differentiate between distinct instances in multi-person images.

## 4.3 ABLATION STUDY

In this section, we analyze the effectiveness of our learning strategy and seek to best practice for distillation-based pretraining. We adopt I2I ReID, human parsing and pedestrian detection as representative tasks for investigation. For I2I ReID, we evaluate $Rank1$ on Market1501. For human parsing, we evaluate $mIoU$ on LIP. For pedestrian detection, we evaluate $AP$ on CrowdHuman.

**The Effect of Major Components.** To investigate the most suitable distillation strategy, we conduct ablation experiments on three types of learning objectives. We use PATH as the teacher model and distill it for 100 epochs on the LUP1M dataset. Comparative results are reported in Table 3. We find

Table 3: Investigating the effect of major components of DPAL, including three alignment objectives and dynamic pattern decoder.

| D-PaDE | $\ell_g$ | $\ell_l$ | $\ell_r$ | I2I ReID | Human parsing | Detection |
|--------|----------|----------|----------|----------|---------------|-----------|
| ✓ | ✓ | | | 95.3 | 52.7 | 87.3 |
| ✓ | | ✓ | | 93.1 | 55.7 | 88.4 |
| ✓ | | | ✓ | 92.5 | 53.0 | 86.4 |
| ✓ | ✓ | ✓ | | 94.9 | 55.7 | 88.1 |
| ✓ | | ✓ | ✓ | 93.9 | 55.5 | 88.1 |
| ✓ | ✓ | | ✓ | **95.3** | 53.1 | 86.7 |
| ✓ | ✓ | ✓ | ✓ | 95.2 | **55.9** | **88.7** |
| - | ✓ | ✓ | ✓ | 95.2 | 55.2 | 87.2 |

that $\ell_g$ focuses more on global information, which allows it to outperform both $\ell_l$ and $\ell_r$ in single-person discrimination task (95.3% vs 93.1% vs 92.5%). As for dense prediction task, $\ell_l$ outperforms $\ell_g$ and $\ell_r$ for human parsing (55.7% vs 52.7% vs 53.0%). When three strategies are combined, they achieve strongest generalization ability across three types of downstream tasks. Therefore, we adopt three-level alignment objectives as our default distillation approach. In addition, removing D-PaDe leads to sub-optimal performance, indicating proposed D-PaDe is capable of alleviating the adverse effect of inter-pattern conflict problem.

Table 4: Ablation study on different teacher (%). ViT-Ti/16 models are pretrained with three-level alignment objectives from different teachers on LUP1M for 100 epochs.

| Method | Arch | # imgs | Teacher | I2I ReID | Human Parsing | Pedestrian Detection |
|--------|------|--------|---------|----------|---------------|----------------------|
| HAP-B | ViT-B/16 | 2.1M | - | 95.5 | 54.8 | 89.6 |
| DPAL | ViT-Ti/16 | 1.2M | HAP-B | 94.1 | 56.1 | 87.7 |
| PATH-B | ViT-B/16 | 12M | - | 93.5 | 59.1 | 90.1 |
| DPAL | ViT-Ti/16 | 1.2M | PATH-B | 95.2 | 55.9 | 88.7 |
| Sapiens-L | ViT-L/16 | 300M | - | 89.4 | 34.8 | 89.5 |
| DPAL | ViT-Ti/16 | 1.2M | Sapiens-L | 85.9 | 48.6 | 85.2 |

**The Effect of Teacher Models.** As shown in Table 4, we explore the effects of DPAL on different teacher models. Specifically, we select three teacher models and perform three-level alignment objectives on LUP1M. Notably, DPAL, utilizing a smaller backbone (ViT-Ti/16) and fewer training images (1.2M), achieves superior human parsing $mIoU$ compared to MAE and HAP, with improvements of approximately 12.4% and 11.5%, respectively. Moreover, DPAL surpasses PATH in the I2I ReID task by 1.7%, despite PATH employing a larger model (ViT-B/16) and a significantly larger dataset (12M images). These results demonstrate that DPAL attains enhanced performance while maintaining a more compact model size, indicating its efficiency and strong generalization capability.

## 5 CONCLUSION

In this paper, we propose DPAL, a novel distillation-based pretraining framework for distilling general knowledge from large HVMs into lightweight ones without requiring access to large-scale source pretraining datasets. Specifically, we design a dynamic pattern decoder that adaptively extracts various visual patterns conditioning on input image and pattern queries. Finally, we introduce three-level alignment objectives to maximize the effectiveness of knowledge transfer from the teacher to the student. Extensive experiments on 15 datasets demonstrate that DPAL achieves strong generalization comparable to much larger models while significantly outperforming previous pretraining methods. Notably, DPAL trains on a limited, unlabeled dataset unseen to large HVMs, providing a more accessible and cost-effective approach to developing generalizable models.

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

## A    THE USE OF LARGE LANGUAGE MODELS

In the preparation of this paper, we employed Large Language Models (LLMs) solely as a writing assistance tool for limited text polishing and language refinement. LLMs were not involved in any aspects of research ideation, conceptual development, technical analysis, algorithm design, experimental execution, or result analyses. All scientific contributions, methodological innovations, and intellectual content remain entirely our own.

## B    DISCUSSION

**Limitation.**    The performance of the student modle is influenced by the teacher model used for pretraining. Additionally, due to limitations in computational resource, we have only tested our method on image datasets and downstream tasks. However, our approach is also applicable to human-centric video understanding tasks, which will be explored in our future work.

**Broader Impact.**    As demonstrated in Section 4, our method outperforms existing pretraining methods across various downstream tasks, highlighting the potential of DPAL as a novel distillation-based pretraining paradigm. Moreover, DPAL serves as an efficient knowledge distillation technique that enables the development of compact variants of large HVMs, making them suitable for deployment on resource-constrained edge devices. Additionally, DPAL eliminates the necessity of accessing the teacher model's original pretraining datasets by utilizing a relatively small open-source dataset of approximately 1 million images for pretraining. This pretraining paradigm significantly reduces training costs and enhances the accessibility of DPAL for the research community, thereby broadening its potential applications. Furthermore, the codebase developed in this work is publicly released to promote reproducibility and further advancements in research.

## C    MORE IMPLEMENTATION DETAILS

### C.1    MODEL ARCHITECTURE

**Backbone.**    We conducted experiments on various student backbones and teacher backbones, with the corresponding settings presented in Table 5.

Table 5: Configuration of neural architectures. Both Vision Transformer (ViT-X) and Swin Transformer (Swin-X) are used for investigation.

| Arch | Patch size | Embed dim | Heads | Blocks |
|---|---|---|---|---|
| ViT-Ti | 16 | 192 | 6 | 12 |
| ViT-S | 16 | 384 | 6 | 12 |
| ViT-B | 16 | 768 | 12 | 24 |

| Arch | Patch size | Window size | Embed dim | Heads | Blocks |
|---|---|---|---|---|---|
| Swin-Ti | 4 | 7 | 96 | (3,6,12,24) | (2,2, 6,2) |
| Swin-S | 4 | 7 | 96 | (3,6,12,24) | (2,2,18,2) |
| Swin-B | 4 | 7 | 128 | (4,8,16,32) | (2,2,18,2) |

**Dynamic Pattern Decoder.**    The dynamic pattern decoder comprises a self-attention module, a router module, three experts and a dynamic expert generator. The router is responsible for assigning experts to different visual tokens. The experts specialize in handing specific patterns. The dynamic expert generator produces weights for experts conditioned on the visual tokens and pattern queries.

**Dynamic Expert Generator.**    The dynamic expert generator consists of self-attention, cross-attention, and FFN modules. We design three learnable expert tokens $T_e = [T_e^1, T_e^2, T_e^3]$, which pass through self-attention, cross-attention, and FFN modules to update the parameters of the three experts. In the cross-attention module, representations from backbone are used as the keys and values, while the expert tokens serve as the queries. The cross-attention module ensures that the parameters of the experts are

Table 6: Configurations of pretraining.

| Configuration | Value |
|---|---|
| Batch size | 2048 |
| Optimizer | AdamW |
| Learning rate | 2.5e–4 |
| Learning rate decay | Consine scheduler |
| Weight decay | 0.05 |
| Warmup epochs | 10 |
| Epochs | 100 |
| Image size | $256 \times 128$ |

updated based on the corresponding representations and pattern queries, enabling each expert to selectively focus on the most relevant pattern.

**Router.** We designed a router that dynamically assigns experts to distant visual patterns, thereby decoupling the alignment learning of three visual patterns. We designed a learnable router token $T_r$, using the representations extracted by the backbone as keys and values. The routing token dynamically adjusts the weights of different experts $W_e$ based on different patterns, enabling the model to effectively capture diverse patterns and enhance its performance in complex visual tasks.

## C.2 Pretraining Details

All lightweight HVMs are pretrained using 8 A6000 48G GPUs. We employ the AdamW optimizer(Loshchilov & Hutter, 2017) with an effective batch size of 2048 (i.e., 256 per GPU). As shown in Table 6, each model is pretrained from scratch for 100 epochs. The learning rate is 2.5e-4 and is decayed via Cosine Annealing scheduler(Loshchilov & Hutter, 2016). The single-person image size is $256 \times 128$, while the multi-person image is $256 \times 256$.

## C.3 Finetuning Details

We utilize representative methods from downstream tasks as baselines, subsequently replacing their backbones with our pretrained backbones for finetuning. The list of codebases used for evaluation is presented in Table 7.

Table 7: Implementation codebases and configurations of fine-tuning on 12 datasets.

| Task | Dataset | Codebases | Image size | Learning rate | Epoch |
|---|---|---|---|---|---|
| I2I ReID | Market1501 (Zheng et al., 2015) MSMT17 (Wei et al., 2018) | SOLIDER (Chen et al., 2023) | $256 \times 128$ | 2e-4 | 120 |
| T2I ReID | CUHK-PEDES(Li et al., 2017) ICFG-PEDES(Ding et al., 2021a) | IRRA (Jiang & Ye, 2023) | $384 \times 128$ | 1e-4 | 60 |
| Attribute recognition | PA100(Liu et al., 2017) PETAzs(Deng et al., 2014) | SOLIDER (Chen et al., 2023) | $256 \times 128$ | 1e-4 | 25 |
| Pose estimation | COCO keypoint(Lin et al., 2014) | ViTPose (Xu et al., 2022) | $256 \times 192$ | 5e-4 | 210 |
| Landmark detection | Whole-body COCO(Jin et al., 2020) | ViTPose (Xu et al., 2022) | $256 \times 192$ | 5e-4 | 210 |
| Human parsing | LIP (Liang et al., 2018) | SOLIDER(Chen et al., 2023) | $576 \times 384$ | 7e-4 | 150 |
| Pedestrian detection | CrowdHuman(Shao et al., 2018) | CrowdDet (Chu et al., 2020) | $1400 \times 800$ | 2e-4 | 30 |
| Multiple human parsing | CIHP(Gong et al., 2018) | Cpi-parser (Wang et al., 2024) | $1333 \times 800$ | 2e-2 | 25 |
| Part-level attribute parsing | Fashionpedia(Jia et al., 2020) | KE-RCNN (Wang et al., 2023) | $1024 \times 1024$ | 1e-4 | 32 |

Table 8: The computational cost in pretraining stage, involving training epoch, training time (Hours) and Memory per GPU (GB).

| Setting | ViTKD | MaskedKD | ScaleKD | TinyMiM | Proteus | DPAL |
|---|---|---|---|---|---|---|
| Epochs | 300 | 300 | 200 | 300 | 300 | 100 |
| Time | 41 | 26 | 60 | 15 | 30 | 22 |
| Memory | 10 | 22 | 26 | 24 | 16 | 26 |
| Downstream Tasks | | | | | | |
| I2I Person ReID | 90.5 | 79.7 | 81.6 | 91.6 | 93.6 | 95.2 |
| Human Parsing | 52.0 | 50.9 | 55.6 | 53.0 | 54.3 | 55.9 |
| Pedestrian Detection | 86.6 | 84.2 | 87.4 | 86.4 | 87.6 | 88.7 |

Table 9: Investigating the effect of DPAL on vision-language spatial reasoning task.

| Evaluation on OmniSpatial | | | | | | |
|---|---|---|---|---|---|---|
| Method | Vision Encoder | Avg. | Dynamic Reasoning | Spatial Interaction | Complex Logic | Perspective Taking |
| LLaVA-1.5-7B | CLIP-ViT-L (304M) | 34.97 | **54.46**/31.23 | 35.29/**36.19**/33.94 | **29.01**/24.08 | **55.60**/34.66/35.14 |
| | DPAL-ViT-T (5M) | **35.62** | 45.95/26.30 | **56.47**/35.24/**42.73** | 18.56/**25.16** | 53.92/**36.70**/**46.99** |

| Evaluation on Robo2VLM | | | | | |
|---|---|---|---|---|---|
| Method | Vision Encoder | Avg. | Spatial Reasoning RS/OS/SR/SU/MV | Goal Reasoning TS-G/TS-S/TS-GL | Interaction Reasoning AU/IP/TU |
| LLaVA-1.5-7B | CLIP-ViT-L (304M) | 21.58 | 35.32/23.87/16.08/**17.78**/17.50 | 31.82/23.79/19.03 | 20.30/**21.74**/22.37 |
| | DPAL-ViT-T (5M) | **24.58** | **53.66**/**27.16**/**19.29**/2.52/17.31 | 29.55/19.88/**46.88** | 17.24/21.18/**31.82** |

Table 10: Investigating the effect of DPAL on vision-language robot control task.

| Simulation Task | ACT (ResNet-18-11M) Easy/Hard | RDT (SigLip-400M) Easy/Hard | PI0 (SigLip-400M) Easy/Hard | H-RDT (SigLip-400M) Easy/Hard | Ours (ViT-Ti-5M) Easy/Hard |
|---|---|---|---|---|---|
| Grab Roller | 66.0/6.0 | 74.0/43.0 | **96.0**/80.0 | 95.0/52.0 | 83.0/**57.0** |
| Place_object_basket | 0.0/0.0 | 42.0/14.0 | 62.0/10.0 | **62.0**/**19.0** | 32.0/4.0 |

# D  ABLATION STUDY

## D.1  TRAINING EFFICIENCY

From the results listed in Table 8, we observe that the proposed DPAL achieves superior downstream performance while maintaining comparable pretraining costs. As shown in Table 1, all methods require 15 40 hours, while the proposed DPAL requires 22 hours. Second, from the perspective of downstream fine-tuning, we choose a fair and widely-used setting, where only the pre-trained backbone is retained in downstream evaluation, and the alignment module is discarded. In this way, the training costs in downstream tasks are the same for all pre-training methods. Based on this, the DPAL does not bring significant computational burden.

## D.2  ABLATION STUDY ON SCALE OF THE DATASET

To explore the optimal scale of the dataset, we construct five subsets of varying scales (0.2M, 0.5M, 1M, 2M, and 4M samples) from the LUPerson dataset for pretraining. As shown in Figure 4, we observe that the performance on the 0.2M and 0.5M subsets is significantly worse than on the 1M subset. Moreover, increasing the dataset size does not lead to further performance improvement. Therefore, LUP1M, as the subset of LUPerson, is sufficient to support distillation-based pretraining.

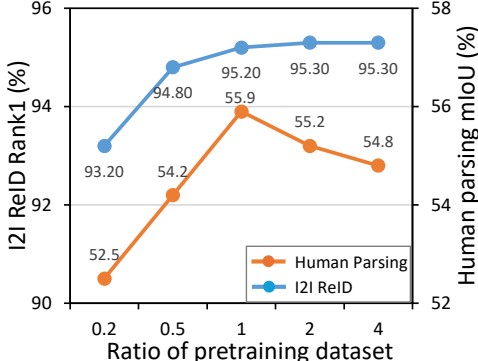

Figure 4: The impact of different scale of datasets on I2I Reid and human parsing tasks.

## D.3  VISION-LANGUAGE TASKS

In this section, we further investigate the effectiveness of proposed method on two vision-language tasks: 1) spatial reasoning; and 2) embodied robot control. Specifically, we choose the InternViT-400M (Chen et al., 2025c) as the teacher model, and use proposed DPAL to distill it into ViT-Ti by leveraging the ImageNet-1M (Deng et al., 2009) as a medium. When testing pretrained model on spatial reasoning task, we replace the vision part of LLaVA-1.5-vicuna-7B (Liu et al., 2023) with the pretrained ViT-Ti, and test it using OmniSpatial (Jia et al., 2025) and Robo2VLM (Chen et al., 2025a) benchmark. As

for the embodied robot control task, we replace the vision part of the H-RDT (Bi et al., 2025) and test it using RobotWin2.0 (Chen et al., 2025b) simulation benchmark. The results reported in Table 9 and Table 10 shows that adopting pretrained lightweight ViT (5M) as the vision encoder achieves competitive performance, which is comparable to that of large vision encoder (400M).

### D.4 ABLATION STUDY ON VARIANTS OF PATTERN DECODER

Pattern decoder functions as an adapter to align the outputs of lightweight HVMs to that of large HVMs. This section study the variants of pattern decoder: 1) MAE-Style (He et al., 2022) decoder, which contains two transformer blocks; 2) Standard MoE, where experts in MoE block is the fixed MLP; and 3) proposed D-PaDe, where the experts are dynamically generated via pattern queries with input image. Comparison results reported in Table 11 show that D-PaDe is the best choice for distillation-based pretraining by far.

Table 11: Ablation study on variants of pattern decoder (%). Aligning ViT-Ti/16 with PATH-B using MAE-style decoder, Standard MoE or D-PaDe.

| Setting | I2I ReID | Human Parsing | Detection |
|---|---|---|---|
| w/o decoder | 95.2 | 55.2 | 87.2 |
| MAE-style | 95.1 | 55.0 | 87.8 |
| Standard MoE | 94.1 | 55.9 | 88.5 |
| D-PaDe | 95.2 | 55.9 | 88.7 |

### D.5 ABLATION STUDY ON VARIANTS OF STUDENT MODEL

We evaluate the performance of DPAL on downstream tasks with different model architectures. We use PATH-B as the teacher model and perform distillation for 100 epochs by default. As shown in the Table 12, we employ vision transformer (Dosovitskiy et al., 2020) for the ViT architecture and swin transformer (Liu et al., 2021b) for the hybrid architecture. The other settings are the same as in Section 4.1 ans Section 4.2. We observe a consistent improvement in model performance concomitant with the increasing model parameters, as exemplified by I2I ReID task where $Rank1$ increases by 0.6% (+16M), 1.2% (+22M), 1.7% (+44M) compared to ViT-Tiny. However, this improvement is accompanied by a corresponding increase in training costs. Moreover, our method is model-agnostic, demonstrating strong performance on both ViT and hybrid architectures.

### D.6 ABLATION STUDY ON MODEL SIZE OF TEACHER

We investigate whether employing teacher models with larger size enhances the performance of the student model. Specifically, we employ PATH-B and PATH-L as teacher models to distill ViT-Tiny. The results presented in Table 13 indicate that increasing the size of the teacher model does not yield performance gains across a wide range of downsream tasks. This may be due to the larger gap between the larger teacher models and the student model, which is also mentioned in the TinyMIM(Ren et al., 2023).

## E VISUALIZATION RESULTS

We provide additional visualization results in Figure 5. First, we visualize the class token representation space of PATH (Tang et al., 2023), Proteus(Zhang et al., 2025), TinyMIM(Ren et al., 2023) and DPAL for two single-person images to investigate the models' ability to learn global identity patterns. As shown in Figure 5 (a), DPAL and PATH distinctly separate the two instances in the representation space, whereas the other methods do not. Second, we conduct principal component analysis (PCA) visualization to investigate the model's capability in capturing local shape patterns. DPAL successfully captures local body shape patterns comparable to those of PATH, while the others fail to capture the whole structure of a person instance. Third, we perform PCA visualization on multi-person images. Similar to PATH, DPAL is able to distinguish different individuals that are depicted by different colors in the visualization. This demonstrates that DAPL has successfully enabled lightweight model to acquire multi-person interaction patterns.

Table 12: Impact of model architecture. We employ PATH-B as teacher model and perform distillation with DPAL on four student architectures.

(a) Single-person discrimitive tasks (%).

| Arch | Type | #Param | I2I ReID | | T2I ReID | | Attribute recognition | |
|------|------|--------|----------|----------|----------|----------|----------|----------|
| | | | Market↑ | MSMT17↑ | CUHK↑ | ICFG↑ | PA100K↑ | PETAzs↑ |
| ViT-Ti/16 | ViT | 5M | 95.2 | 84.3 | 64.3 | 56.0 | 82.4 | 74.0 |
| ViT-S/16 | ViT | 21M | 95.8 | 86.1 | 65.8 | 58.5 | 83.9 | 74.1 |
| Swin-Ti/4 | Hybrid | 27M | 96.4 | 86.2 | 66.9 | 58.5 | 83.1 | 74.9 |
| Swin-S/4 | Hybrid | 49M | **96.9** | **88.2** | **69.6** | **60.0** | **85.9** | **77.1** |

(b) Single-person dense prediction tasks (%).

| Arch | Type | #Param | Pose estimation | | Landmark detection | | Human parsing | |
|------|------|--------|-----------------|----------|--------------------|----------|----------------|----------|
| | | | $AP\uparrow$ | $AR\uparrow$ | $AP\uparrow$ | $AR\uparrow$ | $mIoU\uparrow$ | $mAcc\uparrow$ |
| ViT-Ti/16 | ViT | 5M | 72.6 | 75.8 | 48.8 | 61.5 | 55.9 | 66.7 |
| ViT-S/16 | ViT | 21M | 73.3 | 76.3 | 53.1 | 65.5 | 58.1 | 68.7 |
| Swin-Ti/4 | Hybrid | 27M | 75.1 | 78.1 | 53.9 | 65.7 | 59.3 | 69.7 |
| Swin-S/4 | Hybrid | 49M | **76.3** | **79.4** | **55.6** | **67.2** | **60.7** | **71.5** |

(c) Multi-person visual understanding tasks (%).

| Arch | Type | #Param | Pedestrian detection | | Multiple human parsing | | Part-level attribute parsing | |
|------|------|--------|---------------------|----------|------------------------|----------|------------------------------|----------|
| | | | $AP\uparrow$ | $MR\downarrow$ | $mIoU\uparrow$ | $AP_p\uparrow$ | $AP^{box}_{IoU+F_1}\uparrow$ | $AP^{segm}_{IoU+F_1}\uparrow$ |
| ViT-Ti/16 | ViT | 5M | 88.7 | 45.5 | 51.9 | 50.3 | 39.8 | 37.0 |
| ViT-S/16 | ViT | 21M | 89.2 | 42.9 | **55.9** | **53.4** | 42.9 | 39.3 |
| Swin-Ti/4 | Hybrid | 27M | **90.2** | **42.3** | 55.8 | 53.3 | 42.9 | 39.8 |
| Swin-S/4 | Hybrid | 49M | 89.7 | 43.1 | 55.2 | 52.4 | **44.9** | **41.1** |

(d) Cross-domain perception tasks (%).

| Arch | Type | #Param | Humanart | | Chimpact-Pose | | AP-10K | |
|------|------|--------|----------|----------|---------------|----------|--------|----------|
| | | | $AP\uparrow$ | $AR\uparrow$ | $AP\uparrow$ | $AR\uparrow$ | $AP\uparrow$ | $AR\uparrow$ |
| ViT-Ti/16 | ViT | 5M | 69.9 | 73.4 | 21.9 | 25.5 | 67.0 | 70.3 |
| ViT-S/16 | ViT | 21M | 72.0 | 75.5 | 24.7 | 28.4 | 69.0 | 72.4 |
| Swin-Ti/4 | Hybrid | 27M | 72.9 | 76.1 | 25.2 | 29.4 | 69.4 | 73.0 |
| Swin-S/4 | Hybrid | 49M | **75.1** | **78.3** | **27.8** | **32.1** | **71.5** | **74.7** |

Table 13: Impact of teacher size. We use ViT-Tiny as the student model and perform DPAL distillation separately with teacher models of two different sizes.

(a) Single-person discrimitive tasks (%).

| Teacher | I2I ReID | | T2I ReID | | Attribute recognition | |
|---------|----------|----------|----------|----------|----------------------|----------|
| | Market↑ | MSMT17↑ | CUHK↑ | ICFG↑ | PA100K↑ | PETAzs↑ |
| PATH-B | 95.2 | **84.3** | 64.3 | **56.0** | 82.4 | 74.0 |
| PATH-L | **95.2** | 83.7 | **66.0** | 55.9 | **82.7** | **74.3** |

(b) Single-person dense prediction tasks (%).

| Teacher | Pose estimation | | Landmark detection | | Human parsing | |
|---------|-----------------|----------|--------------------|----------|----------------|----------|
| | $AP\uparrow$ | $AR\uparrow$ | $AP\uparrow$ | $AR\uparrow$ | $mIoU\uparrow$ | $mAcc\uparrow$ |
| PATH-B | 72.6 | 75.8 | **48.8** | **61.5** | **55.9** | **66.7** |
| PATH-L | **72.7** | **78.2** | 48.6 | 61.2 | 55.7 | 66.6 |

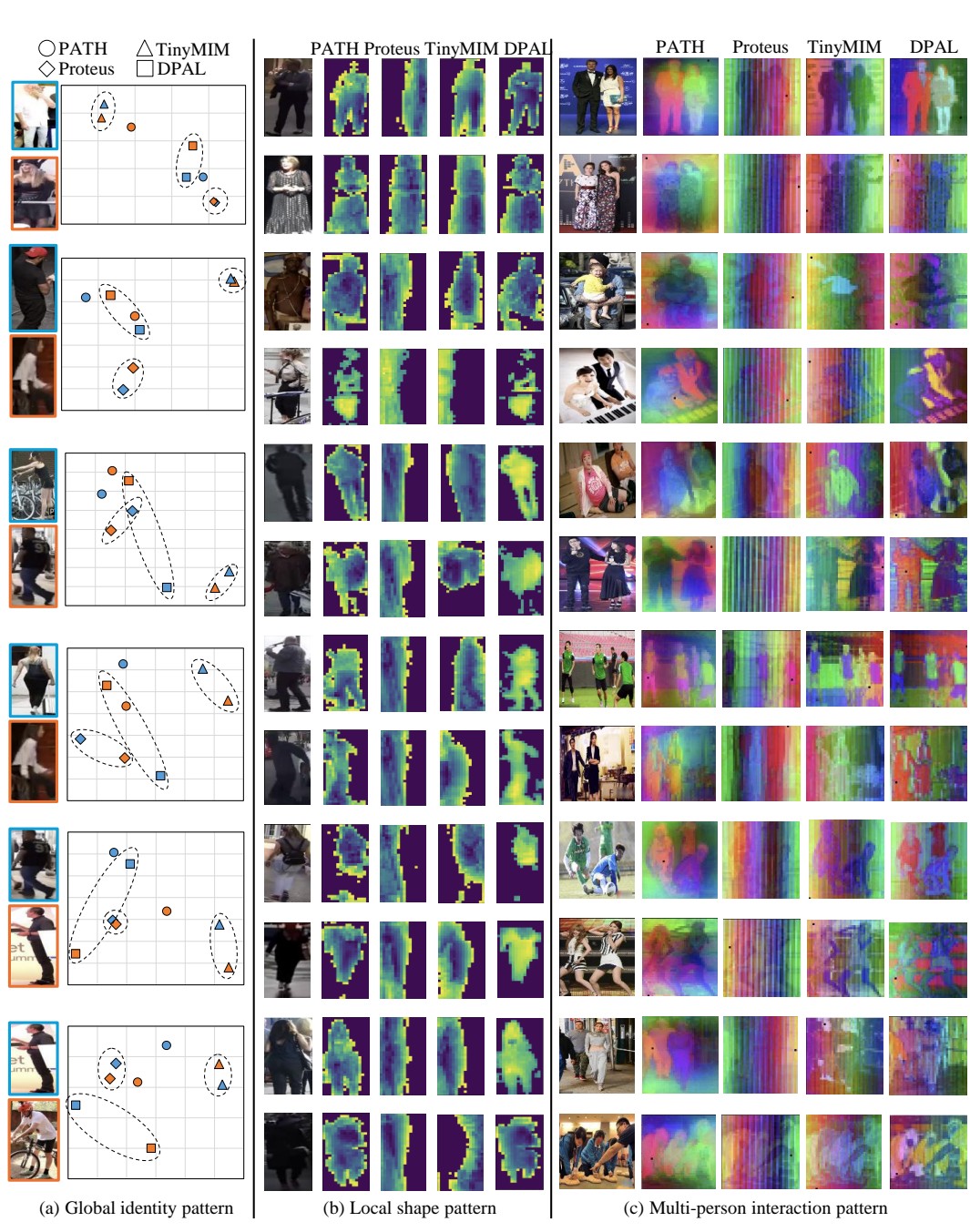

Figure 5: Visualization of learned patterns among four models. From left to right: (a) global identity pattern, (b) local shape pattern, and (c) multi-person interaction pattern.

