# OpenReview forum: "Dynamic Pattern Alignment Learning for Pretraining Lightweight Human-Centric Vision Models"
_ICLR.cc/2026/Conference — ICLR 2026 Conference Withdrawn Submission_

### Official Review · Reviewer_BJ7c · 2025-10-28

**Soundness:** 3
**Presentation:** 3
**Contribution:** 3
**Rating:** 6
**Confidence:** 3

**Summary:**

This paper addresses a critical challenge in the field of human-centric vision: the heavy reliance of state-of-the-art Human-centric Vision Models (HVMs) on large architectures and massive, often inaccessible, pre-training datasets. To democratize the development of powerful HVMs, the authors propose Dynamic Pattern Alignment Learning (DPAL), a distillation-based pre-training framework. The core idea is to transfer the generalization capabilities of a large, pre-trained "teacher" HVM to a lightweight "student" model, using only a small, publicly available, and unlabeled dataset.
DPAL is designed to mimic three key visual patterns essential for human-centric tasks: global identity, local shape, and multi-person interaction. To achieve this without conflicts, the framework introduces a Dynamic Pattern Decoder (D-PaDe), which functions as a dynamic Mixture-of-Experts (MoE) model. This decoder has specialized experts for each pattern, ensuring that they are learned independently. The student model is then trained to align its representations with the teacher's at three levels—global, local, and relational—using these decoded patterns as a medium. The authors demonstrate through extensive experiments on 15 datasets that their lightweight model (DPAL-ViT/Ti) achieves performance comparable to much larger HVMs, significantly outperforming other distillation methods.

**Strengths:**

1. High Practical Relevance and Strong Motivation: The paper tackles a highly significant and practical problem. By enabling the creation of powerful lightweight models without access to proprietary large-scale datasets, it significantly lowers the barrier to entry for both research and real-world deployment.
2. Extremely Comprehensive and Compelling Experimental Results: The empirical evaluation is a major strength of this paper. The authors test their method on an impressive 15 datasets, covering 9 different human-centric tasks as well as cross-domain generalization.
3. Excellent Ablation Studies and Analysis: The paper includes thorough ablation studies that validate the key components of the framework.

**Weaknesses:**

See the questions below.

**Questions:**

1  The paper claims that three experts are conflicted. Why the authors force each visual pattern to be generated independently?

2 I can understand the shape pattern as shown in Fig3 middle, but why is it called local shape? What does ``local’’ mean?

---

### Official Review · Reviewer_tw1Z · 2025-10-28

**Soundness:** 2
**Presentation:** 3
**Contribution:** 2
**Rating:** 4
**Confidence:** 5

**Summary:**

This paper proposes Dynamic Pattern Alignment Learning (DPAL), a knowledge distillation-based pre-training framework designed to efficiently transfer the generalization capabilities of large-scale Human-centric Vision Models (HVMs) to lightweight models without requiring access to the original large-scale pre-training dataset. The core insight is that human-centric vision tasks rely on three heterogeneous visual patterns: global identity patterns (for person re-identification), local shape patterns (for pose estimation), and multi-person interaction patterns (for detection).
The main contributions include:
Dynamic Pattern Decoder (D-PaDe): Designed as a dynamic Mixture of Experts (MoE) model, it incorporates three specialized experts to handle different patterns, effectively avoiding optimization conflicts caused by pattern heterogeneity.
Three-Level Alignment Objective: Aligns representations between the teacher and student models at three levels: global image, local pixel, and instance relationship.
Extensive Experimental Validation: Experiments on 15 datasets demonstrate that ViT-Tiny models with only 5 million parameters, after DPAL pre-training, achieve performance comparable to large-scale models with an order of magnitude more parameters (84 million to 307 million), requiring only 1 million unlabeled images unseen by the teacher model for distillation pre-training.

**Strengths:**

1. The approach of decomposing human-centric vision into three heterogeneous patterns represents a reasonable analysis, while the D-PaDe architecture that dynamically generates experts based on pattern queries is an innovative application.
2. Ablation experiments thoroughly validate the necessity of each component. The evaluation covers 15 challenging datasets, encompassing diverse tasks such as single-person discrimination, dense prediction, multi-person understanding, and cross-domain generalization, making the results highly convincing. Comparisons with multiple baseline methods (including both self-supervised and distillation approaches) and computational efficiency analysis further strengthen the argument.

**Weaknesses:**

1. The description of D-PaDe's specific implementation mechanism lacks precision. Particularly, the process of how pattern queries interact with input features to dynamically generate expert parameters lacks rigorous mathematical formulation, and how the routing module decides which expert to activate is not sufficiently explained.

2. While the authors emphasize "lightweight", relevant analysis is missing. It is well-known that distillation to small models has been extensively validated (e.g., in the DINO series). The proposed distillation method, and further the innovative MoE-based distillation approach, lack analysis regarding the lightweight nature of the MoE model, as well as behavioral analysis of activated experts and different patterns within the MoE framework.

**Questions:**

Pattern Completeness: Are the three proposed visual patterns sufficient to support all human-centric vision tasks? Do other important patterns exist (such as temporal dynamics in videos or fine-grained attribute patterns) that require additional expert modules?

Expert Generation Mechanism: Can you elaborate on how the dynamic expert generator specifically produces expert parameters? How are pattern queries and visual features fused to generate weight matrices? How is the issue of generated experts becoming overly similar or degenerated avoided?

Dataset Dependency: How sensitive is DPAL to the domain of the distillation dataset? If completely general unlabeled images (such as ImageNet or web data) are used instead of human-centric data like LUP1M, would performance significantly decline?

Scalability Analysis: Ablation experiments show that performance does not improve further when the teacher model exceeds PATH-B. Is this purely due to the teacher-student capacity gap? Have attempts been made to use multiple teacher models simultaneously (as in the Theia method) to obtain complementary knowledge?

Routing Design: The current routing uses a simple linear layer. Have more advanced routing mechanisms from the MoE literature (such as top-k routing or load balancing) been explored? What were the results?

Failure Case Analysis: In which specific scenarios or datasets does DPAL underperform? Understanding its limitations and failure modes is crucial for future improvements.

---

### Official Review · Reviewer_7rvU · 2025-10-29

**Soundness:** 2
**Presentation:** 2
**Contribution:** 2
**Rating:** 4
**Confidence:** 3

**Summary:**

This paper introduces Dynamic Pattern Alignment Learning (DPAL), a framework for training small human-centric vision model via distillation from large models (e.g., PATH). The proposed method employs three alignment objectives, each considering different visual patterns (global identity, local shape and instance relation). The patterns are decoded from student representation via Dynamic Pattern Decoder (D-PaDe), a dynamic MoE model with three experts that can help avoiding optimization conflict between different alignment objectives. Empirical evalutaion of DPAL under 15 datasets show that the distilled DPAL-ViT/Ti outperforms self-supervised pretraining or distillation baselines.

**Strengths:**

- **Practical Research Goal:** The method tackles the important and practical challenge of creating generalizable, lightweight Human-centric Vision Models (HVMs) without relying on massive, private pretraining datasets.
- **Novel Distillation Framework:** The paper proposes a novel approach (DPAL) to distill three distinct visual patterns (global, local, and interaction). The core component, a dynamic pattern decoder (D-PaDe), is designed as a dynamic Mixture-of-Experts to mitigate optimization conflicts between these heterogeneous patterns.
- **Strong Empirical Results:** The lightweight 5M-parameter model shows impressive performance across 15 diverse datasets, significantly outperforming prior distillation methods.

**Weaknesses:**

- **Ambiguous 'Generalization' Claim:** The claim of distilling "generalization capability" is potentially overstated. The framework's design appears to be a form of multi-task distillation with objectives that are highly tailored to the downstream evaluation categories, rather than a method for learning a truly general representation.
    - For instance, the global-level objective (aligning multiple student representation from multi-view images to single teacher representation) and the local-level objective (using a teacher-generated body mask) introduce strong inductive biases that directly benefit the ReID and dense prediction tasks, respectively.
    - This strong alignment between objectives and evaluation tasks makes it difficult to assess if the model has learned an emergent, general ability or has simply been co-designed to excel on this specific benchmark.
- **Clarity and Presentation Issues:** The paper's clarity is hindered by several issues that affect understanding and reproducibility:
    - **Unconventional Terminology:** The paper uses non-standard terms. For example, referring to an $L \times L$ attention matrix as an "attention token" seems unconventional. More fundamentally, the term "distillation-based pretraining" is confusing; this process can be simply described as knowledge distillation.
    - **Undefined Notations:** Some notations are used without proper definition. For example, the teacher-side targets (e.g., $\widetilde{F_{t}^{g}}$ and $\widetilde{F_{t}^{l}}$) in Equations (4) and (5) are used without a clear definition in the text.
    - **Missing Visualization Details:** The methodology for generating the qualitative visualizations in Figure 3 is not explained in the paper. This makes the accompanying qualitative analysis difficult to interpret, as it's unclear how the visualizations support the claim about instance distinction (”DPAL can distinguish between different instances”). It is also hard to interpret what it means by “achieves results comparable to teacher model” (L421).
- **Weak Empirical Support for D-PaDe:** The central claim that the *dynamic* nature of D-PaDe is crucial for mitigating “inter-pattern conflict” is not sufficiently supported.
    - **Insufficient Ablation for 'Dynamic' Design:** The main paper (Table 3) lacks a critical comparison against static decoder baselines. While an ablation exists in the appendix (Table 11), the results are not compelling, showing marginal or no gains for D-PaDe over the baselines. This minimal improvement does not seem to strongly justify the added complexity of the dynamic expert generator.
    - **Unsupported Claim of 'Conflict Mitigation':** The paper's only evidence for conflict mitigation is that removing D-PaDe entirely hurts performance, which is insufficient. A more rigorous experiment would be to repeat the loss-component ablation (e.g., $l_g$ only, $l_l$ only, …) on the "w/o D-PaDe" baseline. If D-PaDe truly mitigates conflict, one would expect to see harsher performance trade-offs on the ablated model, which is not demonstrated.

**Questions:**

- **Justification for Specialized HVMs vs. General-Purpose Baselines:** I would like to hear from the authors about the high-level motivation for HVMs, as I am not much familiar with HVMs.
    - What is the core justification for pretraining specialized human-centric models, as opposed to simply fine-tuning general-purpose vision foundation models for these tasks?
    - Could the authors conduct experiments for comparison to a simple baseline using an lightweight model trained on natural image (e.g., DINO-S), after simply fine-tuning it on human dataset? This comparison seems essential to validate that the complex distillation from a large HVM provides a tangible benefit over standard pretraining.
    - For the self-supervised baselines, were these trained from scratch on LUP1M, or were they initialized from some general-domain checkpoints? I suspect they can perform better if initialized from general-domain checkpoint as the scale of dataset is limited.
- **Clarification on Self-Attention Operation:** There is an ambiguity in the equations (1-3).
    - Equation (1) shows $SA(F_{s1}^{g})$, where $F_{s1}^{g}$ is defined as the single class token. Applying standard self-attention to a single token would be mathematically redundant and equivalent to a linear projection. Could the authors clarify the implementation?
    - This formulation also appears to conflict with the description in Section 3.1 ("a self-attention module SA() is used to encode pattern queries" ) and the diagram in Figure 2, which implies SA is applied to entire sequence of feature tokens, not only on single token. Please clarify this discrepancy.
- **Ablation on Cross-Domain Tasks:** The ablation study in Table 3 only uses three representative tasks. Given the concerns about task-specific inductive biases (e.g., the foreground mask), it would be insightful to see this ablation extended to a cross-domain task (e.g., HumanArt). This would help test whether all three alignment objectives are truly beneficial in a more general setting where the injected biases may not be as directly applicable.

Minor suggestions

- It would be helpful to include the teacher model's performance in the main result tables. This would allow readers to easily assess the distillation gap for each task.
- L469 mentions improvements of 12.4% and 11.5% over MAE and HAP, respectively. However, it seems there is no corresponding values in Table 4.

**Details Of Ethics Concerns:**

The paper's pretraining relies on the LUP1M dataset, a subset of LUPerson. This dataset was created by scraping streetview videos in YouTube, which uses images of unconsented human subjects (Privacy) and potentially have copyright or crawling policy issues (Legal Compliance). The paper does not have ethics statement to justify those concerns.

---

### Official Review · Reviewer_nh4g · 2025-10-31

**Soundness:** 2
**Presentation:** 3
**Contribution:** 2
**Rating:** 4
**Confidence:** 4

**Summary:**

This paper proposes Dynamic Pattern Alignment Learning (DPAL), a distillation-based pretraining framework to transfer the generalization ability of large human-centric vision models (HVMs) to lightweight ones. DPAL decomposes supervision into three heterogeneous “visual patterns”—global identity, local shape, and multi-person interaction—via a Dynamic Pattern Decoder (D-PaDe) that behaves like a dynamic MoE to generate pattern-specific representations for alignment. Training uses three losses (global MSE over multi-views, local MSE on patch tokens, and relation-level KL on instance interactions). The student encoder is kept for downstream tasks while the teacher and D-PaDe are discarded. Experiments span 15 datasets across ReID, attributes, pose, parsing, detection and cross-domain generalization; the authors claim the 5M-parameter ViT-Tiny student approaches PATH-B/Sapiens-L and surpasses distillation baselines (Proteus, TinyMIM), using ~1M unlabeled images (LUP1M) plus copy-paste synthesis for multi-person patterns.

**Strengths:**

- Clear factorization of supervision into global/local/relation levels, with explicit objectives and a unifying D-PaDe adapter; losses are simple and well-defined.
- Task-agnostic pretraining that aims for broad transfer across 15 human-centric benchmarks and even cross-species/style settings (HumanArt, AP-10K, Chimpact-Pose).
- Practical deployment story: D-PaDe is only used in pretraining; the student backbone is retained for fine-tuning/inference.
- Accessible data recipe (LUP1M) and a stated plan to release code/checkpoints.

**Weaknesses:**

- Reliance on synthetic multi-person composition. Multi-person “interaction” supervision is created by simple copy-paste, which may distort occlusion statistics, scale coherence, and contextual cues that real crowd datasets exhibit. Please quantify domain gap vs. real multi-person corpora (e.g., CrowdHuman) with a pretraining ablation without any synthetic composites and with only real multi-person images, holding compute constant.
- Unclear computational overhead & stability of D-PaDe. D-PaDe is described as a dynamic MoE that generates experts conditioned on tokens/queries and “activates one expert,” but the paper lacks parameterization details, FLOPs/latency overhead during pretraining, initialization, and training stability (e.g., collapse/degenerate routing). Provide expert generator architecture, parameter counts attributable to D-PaDe, wall-clock costs, and convergence plots; compare against a non-dynamic adapter with similar capacity.
- Teacher dependence and fairness. Many claims hinge on the teacher choice (PATH-B vs. Sapiens-L). Please provide a systematic study showing (a) performance vs. different teachers at similar effective capacity, (b) whether local/relation losses still help when the teacher lacks attention tokens (or if attention is noisy), and (c) whether results remain competitive when the teacher is weaker or mismatched. Currently, the method may be over-tuned to strong ViT teachers with rich attention signals.
- Ablation insufficiency on loss balancing. All loss weights are set to 1 for simplicity, but this hides sensitivity.

**Questions:**

See Weakness.

---

### Note · Authors · 2025-12-17

I have read and agree with the venue's withdrawal policy on behalf of myself and my co-authors.